# Novel dual HIV maintenance therapy with nevirapine plus lamivudine retain viral suppression through 144 weeks—A proof-of-concept study

C. R. Kahlert[1,2]*, M. Cipriani[1], P. Vernazza[1]

1 Infectious Diseases and Hospital Epidemiology, Cantonal Hospital Sankt Gallen, St. Gallen, Switzerland,
2 Infectious Diseases and Hospital Epidemiology, Children's Hospital of Eastern Switzerland, St. Gallen, Switzerland

* christian.kahlert@kssg.ch

**Data Availability Statement:** The minimal data set underlying the study is available in a data repository with the address: https://doi.org/10.5281/zenodo.3974105.

## Abstract

### Objectives

The aim of this proof-of-concept study is to test feasibility and efficacy of NVP plus Lamivudine (3TC) as novel simplified HIV maintenance dual therapy (DT) strategy.

### Methods

Patients under combined antiretroviral treatment (cART) with fully suppressed HIV plasma viral load (pVL) >24 months–whereof >6 months on an NVP- containing regimen—were switched to oral NVP plus 3TC for 24 weeks. Patients could then decide whether to continue DT or return to the previous cART. HIV pVL was monitored monthly until week 144. The primary outcome was confirmed viral failure (RNA >100 copies/ml). Low-level detection of HIV-RNA in plasma was compared in each patient with pre-study viral load measurements.

### Results

Twenty patients were included, switched to DT and all completed week 24. One patient decided thereafter to discontinue study participation for personal reasons. After a total of 144 observation weeks, none of the patients failed. The frequency of low- level HIV-RNA detection was not different from the period before randomization.

### Conclusions

Our findings are surprising but given the nature of a proof-of-concept study, the results do not support the use of this dual regimen. However, as this dual HIV maintenance strategy was feasible and effective, over a period of 144 weeks, we suggest NVP plus 3TC warrants further evaluation as potential maintenance option in patients tolerating nevirapine. A properly sized multicentre non-inferiority trial is ongoing to further evaluate the value of this DT maintenance strategy.

**Funding:** The author(s) received no specific funding for this work.

**Competing interests:** The authors have declared that no competing interests exist.

## Introduction

Today, combination antiretroviral therapy (cART) is recommended for all patients with HIV infection [1]. Lifelong cART demand from the patient a long breath by daily regular intake of medication and is associated with potential long-term adverse events and high cost. The combination of several antiretroviral compounds is essential at the start of treatment, as high viral replication can allow rapid development of resistance. Most of currently recommended cART regimen for treatment induction, quickly result in undetectable HIV plasma viral load (pVL). Afterwards, viral replication is completely suppressed and a simplified treatment with two antiretroviral compounds (dual therapy–DT) or even one substance (monotherapy) may hypothetically control viral replication on the long run. The underlying hypothesis of a simplified treatment with two antiretroviral compounds (dual therapy–DT) or even one substance (monotherapy) is, that low viral replication is associated with a reduced risk of viral escape and thus may control viral replication on the long run. Moreover, less drugs translates into lower risk of long-term adverse events and reduced costs. However, most monotherapy trials (MT) showed elevated occurrence of a short-term increase in HIV pVL >50–200 copies/ml that is followed by HIV viral suppressions and is considered a viral "blip" [2]. As a possible explanation, some MT demonstrated evidence for a decreased antiretroviral activity in the central nervous system [3, 4].

Nevirapine (NVP), a non-nucleoside reverse-transcriptase-inhibitor (NNRTI) and lamivudine (3TC) a nucleoside reverse-transcriptase-inhibitor (NRTI) are two drugs that are known for more than 20 years in clinical practice with documented long-term tolerance and efficacy over decades [5–7]. They are available as generic drugs and thus represent a cost saving strategy. For both compounds, compartment penetration into sanctuaries including the central nervous system is exceptional [8]. Interestingly, a recent study evaluating the decrease of the HIV-DNA reservoir in HIV infected individuals under cART identified NVP therapy as a distinct factor reducing the viral reservoir [9]. Further studies are needed to understand to what extent the outstanding compartment penetration of NVP explains this hitherto unknown advantage of NVP therapy.

Though, clinical use of NVP has declined in recent years and it is currently no longer listed in international guidelines for high-income countries because of a potentially severe adverse drug reaction hypersensitivity reaction (HSR) with maculopapular rash and/or elevated liver enzymes in 7% of patients [10]. When NVP is used in the induction of antiretroviral therapy, about 1% develop potentially lethal reactions including Stevens-Johnson and Lyell-syndromes [11]. HSR to NVP predominantly occurs during the first 12 weeks after start of the drug with a median time from NVP-start to HSR of 30 days (interquartile range [IQR] 17–60) [10]. Although a genetic predisposition to HSR has been suspected, no genetic variants have been consistently associated with the disease so far and currently, it is impossible to absolutely predict NPV-HSR. Therefore, initial close follow-up ensures early detection of potential side effects and is crucial to avoid irreversible adverse drug reactions in patients.

Several risk factors for HSR have been identified in particular a high HIV plasma viral load (pVL) and for this reason NVP is not recommended during the induction of antiretroviral treatment. Based on evidence in over 5000 patients on NVP from three cohorts [12, 13], the situation seems different in the setting of maintenance with a fully suppressed HIV pVL. Accordingly, once HIV-RNA is fully suppressed, the risk of HSR after switching to a NVP-based regimen is lower, and independent of the CD4-count at initiation of NVP. In a recent analysis of this strategy in our clinic, we demonstrated that for NVP-based therapies with a duration of >90 days (n = 221), the overall discontinuation rate was 5.4/100 person years (95% confidence interval [CI] 4.0–7.2) [14]. Combining the above-mentioned benefits of DT and

NVP, patients who are stably supressed for more than 6 months on an NVP-based therapy could profit from a combination of NVP and 3TC. Thus, for this study, we hypothesized that NVP-containing DT with 3TC may provide an optimal HIV maintenance therapy. The aim of this proof of concept study is to test the feasibility and efficacy of this strategy before evaluation in a larger trial.

## Methods

Patients on a stable NVP-containing regimen (>6 months) with a fully suppressed HIV pVL (≤ 50 copies/ml) for more than 24 months were switched to DT with oral NVP (400mg/d) and 3TC (300mg/d). Both compounds were administered once daily with or without food. HIV pVL was monitored monthly until week 24 (COBAS® TaqMan® 48 Analyzer by Roche Diagnostics). The primary outcome was confirmed viral failure (RNA >100 copies/ml) within 24 weeks. To limit the potential harm to patients in particular to prevent the development of resistance in the event of virus failure, a stopping rule was implemented. The study would have been terminated prematurely if two patients would have demonstrated viral failure. In addition, the protocol required a stepwise inclusion: After the five first inclusions further inclusion of five more patients was initiated after all five first patients have reached week 12 without viral failure. When 10 patients reached week 12, the remaining 10 patients were included. After week 24, patients were offered to be further continued on DT and monitored every 8 weeks for another 24 weeks. Subsequently, monitoring was reduced to every 12 weeks until week 96 and every 24 weeks until week 144. The frequency of low-level detection of HIV pVL (<20, 20–50, >50 copies/ml) was compared in each patient with pre-study viral load measurements. Since all patients were also participants of the Swiss HIV Cohort Study (SHCS), treatment adherence was assessed with the self-report instrument used in the SHCS every 24 weeks. Descriptive statistics were used to characterize the study population. This study was approved by the cantonal ethics committee (Ethikkommission Ostschweiz, EKOS, ID: EKOS 2016–01963) and registered (ClinicalTrials.gov Identifier: NCT03223402). All participants provided written informed consent.

## Results

Recruitment was initiated in December 2016 and all patients were recruited over a period of 6 months. Twenty patients were included and switched to DT with 3TC and NVP (Fig 1). At baseline, all participants had undetectable HIV pVL. All reached the 24-week primary endpoint. 19/20 (95%) patients decided to continue on DT. One patient decided not to continue the observation period for personal reasons.

We here present the complete follow up until week 144. Patient characteristics are presented in Table 1.

Patients were generally pre-treated for a long time, with a median time on cART of 7.5 years (IQR 6–12). One patient had chronic hepatitis B (HBV) co-infection with suppressed HBV-DNA on a tenofovir-containing cART. However, the patient insisted on discontinuing tenofovir because of personal concern about tubular side effects and suspected inhibition of telomerase activity by this drug. In addition, the patient was highly motivated to participate in this study, for which reason the principal investigator made an exception to the exclusion criterion. After switch to 3TC-containing DT, this patient still showed a negative HBV viral load.

During 144 weeks, no patient failed DT with NVP and 3TC (Fig 2). Of notice, all measurements were <50 copies/ml, except one value of 55 copies/ml at week 96, which resulted to be <20 copies/ml after repetition at week 99. Low-level plasma HIV-RNA detections was rare both during the study and observation period (309 HIV-RNA measurements) and during 144

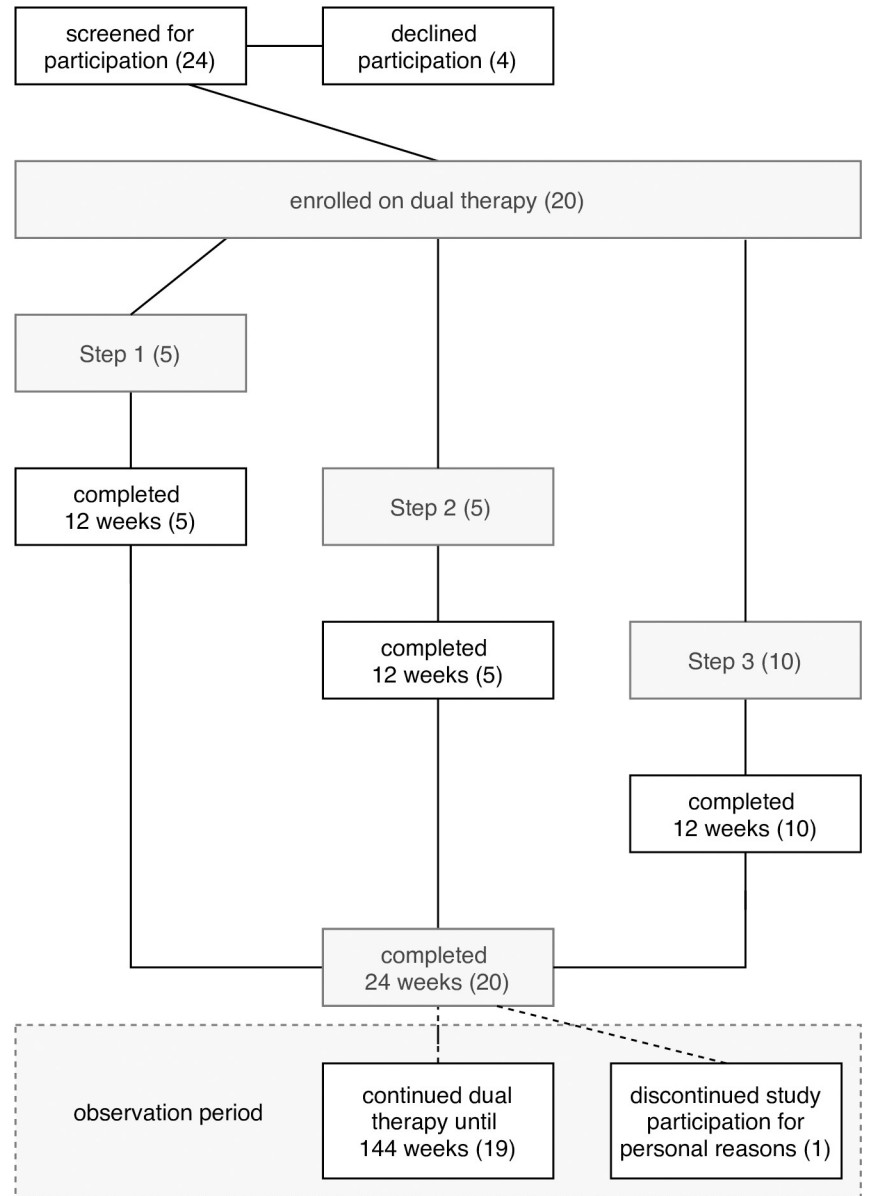

**Fig 1. Flow chart of participant enrolment, follow up and observation period.** Study flow with illustration of stepwise enrolment.

weeks prior to the study, on stable cART (120 measurements). In order to compare the likelihood of blips in both periods, we compared the proportion of measurements with detectable HIV-RNA or HIV-RNA $\geq$ 20 copies/ml before and during the study for every patient and compared these proportions between the two periods with Wilcoxon signed-rank tests. Both comparisons revealed very similar frequencies of blips before and during DT, and no significant change (detectable mean frequeny: 17.1% vs 15.7%, p = 0.43; >20cp/ml: 1.2% vs. 4.2%, p = 0.28).

## Conclusion

In this proof-of-concept study of 20 HIV patients on a stable NVP-containing maintenance DT with 3TC, HIV pVL remained suppressed in all 20 patients for 24 weeks. Furthermore, 19

**Table 1. Patient baseline characteristics.**

| Patients included in the pilot study | 20 (100%) |
|---|---|
| **Age (years)** | |
| mean | 52.4 |
| maximum | 81 |
| minimum | 34 |
| **Gender** | |
| Female | 5 (25%) |
| Male | 15 (75%) |
| **Race** | |
| Caucasian | 20 (100%) |
| **Relevant Comorbidity** | |
| Hepatitis C | 1 (5%) |
| Hepatitis B (HBV-DNA suppressed) | 1 (5%) |
| **Co-Medication** | |
| Antihypertensive | 3 (15%) |
| Statins | 3 (10%) |
| Antithrombotic agents | 2 (10%) |
| Antiplatelet | 1 (5%) |
| **Time on ART (years)** | |
| median | 7.5 |
| maximum | 22 |
| minimum | 3 |
| **HIV-Stage (CDC classification)** | |
| A1 | 2 (10%) |
| A2 | 11 (55%) |
| A3 | 1 (5%) |
| B2 | 1 (5%) |
| B3 | 1 (5%) |
| C3 (2x PcP, 1x TB, 1x esophageal candidiasis) | 4 (20%) |
| **CD4 Nadir (CD4/µl, %)** | |
| mean | 266 (16%) |
| maximum | 580 |
| minimum | 0 |
| *last before switch to DT*: | |
| mean | 680 (38%) |
| maximum | 1170 |
| minimum | 260 |
| **Time on NVP before pilot study (months)** | |
| median | 54.5 |
| maximum | 88 |
| minimum | 6 |
| **Last ART before pilot study** | |
| 3TC ABC NVP | 16 (80%) |
| TDF ETC NVP | 4 (20%) |

**CDC classification** available under: https://www.cdc.gov/mmwr/preview/mmwrhtml/00018871.htm

**ART**: antiretroviral therapy, **NVP**: Nevirapine, **3TC**: Lamivudine, **ABC**: Abacavir, **TDF**: Tenofovir, **ETC**: Emtricitabine **PcP**: Pneumocystis jirovecii pneumonia, **TB**: tuberculosis.

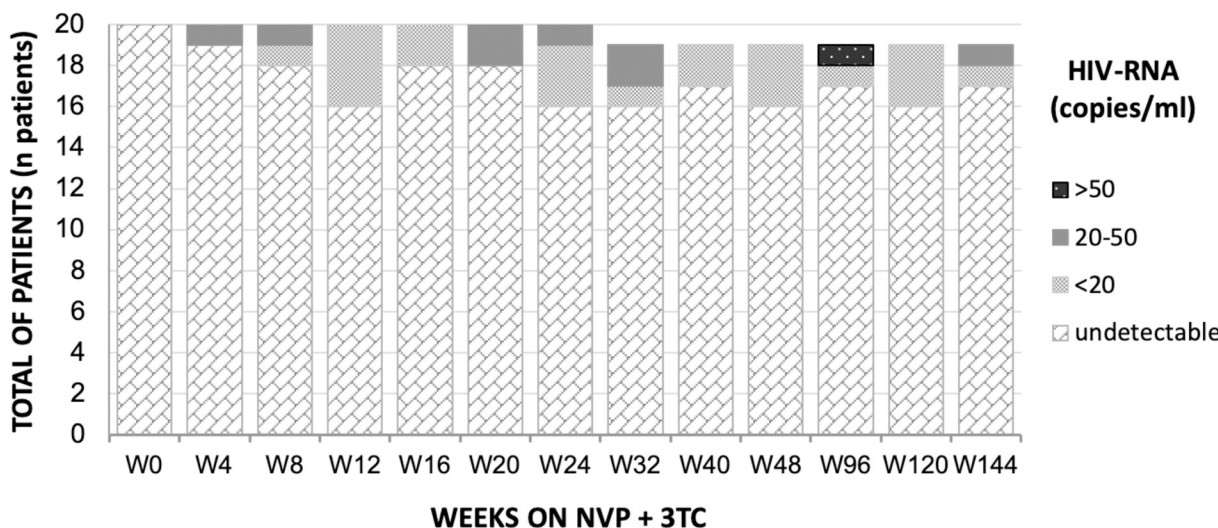

**FREQUENCY OF LOW HIV-RNA DETECTION**
**W0-W144**

The number of patients with HIV RNA results (i) undetectable, (ii) <20 copies/ml, (iii) 20-50 copies/ml or (iv) 50-100 copies/ml is displayed.

**Fig 2. Plasma HIV-RNA during week 0—week 144.** The HIV RNA results is displayed for all participants.

patients who continued DT after the primary endpoint during the observation phase remained stably suppressed until week 144. To the best of our knowledge, these findings have not been described before and thus are of interest as proof of concept. This novel maintenance cART strategy is attractive because effective and safe in selected patients tolerating NVP for more than 6 months and stable viral suppression. As both drugs have been used for more than 20 years, there is extensive experience with the long-term safety in addition to the benefit of cost savings.

Our study has several limitations. This was a small single-centre, non-randomized, single-arm study. All patients were fully suppressed for at least 24 months and tolerated NVP for minimally 6 months. However, in this population, all patients kept full HIV pVL suppression and even the frequency of blips with DT was similar to the pre-study period where patients received cART with 3 compounds. Although this does not guarantee success in a properly sized clinical trial, this is remarkable because NVP has a low resistance barrier and requires only a single mutation in the reverse transcriptase genome. But as stated, this risk is greatest in the beginning of therapy when ongoing viral replication allows the selection of resistant mutants. The observation of a stable rate of viral blips after treatment simplification contrasts with the observation of an increased rate of viral blips in many monotherapy studies, even if compounds with a high barrier against drug resistance have been used [15]. Given the documentation of viral failure of protease monotherapy in the central nervous system [3, 4], we hypothesize that the excellent compartment penetration of NVP and 3TC is an important characteristic of the DT selected in this study. Protease inhibitors and integrase strand transfer inhibitors were both suggested for monotherapy, due to the high genetic barrier but exhibited limited activity as monotherapy. A high penetration into sanctuaries may thus be more important than a high genetic barrier in the context of antiretroviral maintenance. This hypothesis is further supported by the increased correlation of NVP use with undetectable proviral

HIV-DNA [9]. However, this study was unable to prove or rule out low-level resistance in participants with a blip. Our attempt to perform next generation sequencing from the sample with an HIV-RNA value of 55 copies/ml was unsuccessfull.

Today, nevirapine is used less frequently in clinical practice in particular in high-income countries. The main reason is the high risk of hypersensitivity reactions (HSR) at treatment start. An additional reason might also be the lack of any supporting marketing effort. After the demonstration of markedly reduced HSR-rates when initiated with undetectable viral load [10, 12, 13], the marketing efforts to re-introduce the drug into the marked were almost nil. The cost saving potential of the use of generic drugs is immense. Despite the risks and lack of marketing efforts for this "old drug", NVP has been used by some patients for many years and is available as a generic drug in high-income countries. In an evaluation of a strategic decision to use nevirapine as part of standard maintenance cART we recently were able to demonstrate that more than a third of patients could safely be switched to nevirapine [14].

In summary, given the sample size, our findings do not reject the null-hypothesis that dual therapy is equally effective as standard therapy. Still, the results of this proof-of-concept study support further evaluation of this novel dual HIV strategy using NVP plus 3TC as potential maintenance option in HIV infected patients tolerating NVP. As argued above, the advantage is long-term safety and cost. A properly sized multicentre non-inferiority trial is ongoing to further evaluate the value of this DT maintenance strategy.

## Supporting information

**S1 File. TREND statement checklist.**
(PDF)

**S2 File. Study protocol.**
(PDF)

## Acknowledgments

The authors acknowledge the participation of the patients and the helpful support of Patrick Schmid and the whole clinic staff. In addition, we thank Prof. Thomas Klimkait (Department of Biomedicine, Molecular Virology, University of Basel, Switzerland) for the collaboration on performing next generation sequencing on samples showing a blip and Dr. Sabine Güsewell (Clinical Trials Unita Cantonal Hospital St. Gallen, Switzerland) for her statistical support.

## Author Contributions

**Conceptualization:** C. R. Kahlert, P. Vernazza.

**Data curation:** C. R. Kahlert, M. Cipriani, P. Vernazza.

**Formal analysis:** C. R. Kahlert, M. Cipriani, P. Vernazza.

**Investigation:** C. R. Kahlert, M. Cipriani, P. Vernazza.

**Methodology:** C. R. Kahlert, P. Vernazza.

**Supervision:** P. Vernazza.

**Validation:** C. R. Kahlert, M. Cipriani.

**Visualization:** C. R. Kahlert, M. Cipriani, P. Vernazza.

**Writing – original draft:** C. R. Kahlert.

**Writing – review & editing:** C. R. Kahlert, M. Cipriani, P. Vernazza.

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
