## [Decision Letter · Decision Letter 0]

19 Nov 2019

PONE-D-19-28328

NOVEL DUAL HIV MAINTENANCE THERAPY WITH NEVIRAPINE PLUS LAMIVUDINE RETAIN VIRAL SUPPRESSION THROUGH 96 WEEKS - A PROOF-OF-CONCEPT STUDY

PLOS ONE

Dear Dr. Kahlert,

Thank you for submitting your manuscript to PLOS ONE. After careful consideration, we feel that it has merit but does not fully meet PLOS ONE’s publication criteria as it currently stands. Therefore, we invite you to submit a revised version of the manuscript that addresses the points raised during the review process.

We would appreciate receiving your revised manuscript by Jan 03 2020 11:59PM. To enhance the reproducibility of your results, we recommend that if applicable you deposit your laboratory protocols in protocols.io, where a protocol can be assigned its own identifier (DOI) such that it can be cited independently in the future. For instructions see: http://journals.plos.org/plosone/s/submission-guidelines#loc-laboratory-protocols

We look forward to receiving your revised manuscript.

Kind regards,

Alan Winston

Academic Editor

PLOS ONE

Journal Requirements:

This study was sponsored by the Clinics of Infectious Diseases and Hospital Epidemiology, Cantonal Hospital St. Gallen, Switzerland

Reviewers' comments:

Reviewer's Responses to Questions

**Comments to the Author**

1. Is the manuscript technically sound, and do the data support the conclusions?

Reviewer #1: Partly

Reviewer #2: Yes

2. Has the statistical analysis been performed appropriately and rigorously? 

Reviewer #1: Yes

Reviewer #2: N/A

3. Have the authors made all data underlying the findings in their manuscript fully available?

Reviewer #1: Yes

Reviewer #2: Yes

4. Is the manuscript presented in an intelligible fashion and written in standard English?

Reviewer #1: Yes

Reviewer #2: Yes

5. Review Comments to the Author

Reviewer #1: 1) Evidently this is an interesting piece of work that requires further evaluation in a larger study for patients who are currently taking nevirapine.

The paper states that nevirapine is a safe drug to switch to "regardless of CD4 count" in those who are undetectable on ART. I think that there are some significant questions about nevirapine as a stable switch strategy which are not addressed in this paper. The evidence concerning this question is observational and retrospective. The comparator in these papers is rate of HSR in anti-retroviral naive patients with low CD4 counts starting nevirapine not people starting ART containing alternative agents. Indeed in the paper of Wit et al, 2008 for those patient switching to nevirapine in the "high-high" group, the lower limit of the confidence interval for the odds of HSR only just touches 1. This is a drug assoicated with liver toxicity and HSR but with guidelines which limit the risk of these adverse events.

I think it is important to clarify this as, while the ambition to maintain use of older, cost-effective drugs is admirable, readers should be aware of this limitation in the context of the wide availability of other third agents and strategies in the current era. If we are only comparing nevirapine with nevirapine then this strategy has limited scope.

2) I think you might want to just check through the references again - line 181 indicates reference 8 with respect to reduced HSR, but this reference is about CNS PK.

Reviewer #2: Thank you for this interesting and, as you say in your conclusion, surprising study. I think this deserves publication but with some amendments:

1) Abstract conclusion: "Our findings are surprising but do not falsify the null-hypothesis that dual therapy is inferior to standard therapy" - I do not understand this statement. Do you mean to say that "this pilot study, though promising, is not powered to prove whether this therapy option is non-inferior to standard of care" - please revise so clear

2) You abstract conclusion point "having the advantage of reduced adverse events and cost" should be removed. You have not demonstrated, as far as I can see, reduced AE in this small, single arm pilot and in the era of largely generic NRTI there may be no cost advantage - indeed, in England, the difference between 3TC and TDX/FTC generic is less than £1 a week....

3) I think to describe your own findings as 'surprising' you need to better justify your rationale for undertaking the trial and expand on the information provided to participants with respect to risk of failure or resistance development

4) The fact that NVP is not recommended in any high-income country guidelines should be acknowledged; similarly you need to provide some clinical context for recommending a regimen like this - why choose NVP/3TC over more established tripe/dual options?

5) Please include some discussion as to the value of pilots e.g. PADDLE is an example of promising pilot findings being replicated in a larger single-arm study and then large RCTs, PI/r + MVC is an example whereby promising single-arm results were not replicated in RCTs

6. PLOS authors have the option to publish the peer review history of their article (what does this mean?). If published, this will include your full peer review and any attached files.

Reviewer #1: No

Reviewer #2: No

---

## [Author Response · Author response to Decision Letter 0]

3 Dec 2019

***RESPONSE*** to REVIEWER COMMENTS

Reviewer #1

1) Evidently this is an interesting piece of work that requires further evaluation in a larger study for patients who are currently taking nevirapine. The paper states that nevirapine is a safe drug to switch to "regardless of CD4 count" in those who are undetectable on ART. I think that there are some significant questions about nevirapine as a stable switch strategy which are not addressed in this paper. The evidence concerning this question is observational and retrospective. The comparator in these papers is rate of HSR in anti- retroviral naive patients with low CD4 counts starting nevirapine not people starting ART containing alternative agents. Indeed in the paper of Wit et al, 2008 for those patient switching to nevirapine in the "high-high" group, the lower limit of the confidence interval for the odds of HSR only just touches 1. This is a drug assoicated with liver toxicity and HSR but with guidelines which limit the risk of these adverse events. I think it is important to clarify this as, while the ambition to maintain use of older, cost-effective drugs is admirable, readers should be aware of this limitation in the context of the wide availability of other third agents and strategies in the current era. If we are only comparing nevirapine with nevirapine then this strategy has limited scope.

***We thank the reviewer for these constructive and helpful comments. Absolutely, the caution of using Nevirapine must remain clearly articulated. We have adapted the introduction accordingly. We specified that the evidence is based on cohort studies and changed the sentence from “ART can be safely switched” to “the risk of HSR after switching to a NVP-based regimen is lower” (Line 76). At the same time, we would like to emphasize that we do not propose a switch in our manuscript, but instead propose dual therapy (DT) with NVP and 3TC as maintenance for patients who tolerate NVP more than 6 months.***

2) I think you might want to just check through the references again - line 181 indicates reference 8 with respect to reduced HSR, but this reference is about CNS PK.

***We totally agree and thank the reviewer for this comment. The reference was changed from reference 8 to references 10,12 and 13.***

Reviewer #2

Thank you for this interesting and, as you say in your conclusion, surprising study. I think this deserves publication but with some amendments:

1) Abstract conclusion: "Our findings are surprising but do not falsify the null-hypothesis that dual therapy is inferior to standard therapy" - I do not understand this statement. Do you mean to say that "this pilot study, though promising, is not powered to prove whether this therapy option is non-inferior to standard of care" – please revise so clear

***We thank the reviewer for all these valuable comments and suggestions. In this sentence we want to clearly express that the results do not support the use of this DT maintenance therapy due to the small sample size. However, they are surprising and do support the conduct of a properly sized clinical trial. To elucidate this intention, both the summary and the last paragraph of the manuscript have been adapted.***

2) You abstract conclusion point "having the advantage of reduced adverse events and cost" should be removed. You have not demonstrated, as far as I can see, reduced AE in this small, single arm pilot and in the era of largely generic NRTI there may be no cost advantage - indeed, in England, the difference between 3TC and TDX/FTC generic is less than £1 a week....

***You are right, a study design without a control group cannot prove a reduction of adverse events. Therefore, we removed this sentence from the abstract conclusion. However, it is evident that the risk for cART-associated long-term side effects is reduced with a DT versus triple cART. Concerning reduced costs, this of course varies by country. In Switzerland, we currently still do not have generic TDF/FTC that’s why the difference with 3TC is still significant with about £140 a week. With regard to NVP, the cost saving potential is even more than 50% compared to a therapy with e.g. dolutegravir.***

3) I think to describe your own findings as 'surprising' you need to better justify your rationale for undertaking the trial and expand on the information provided to participants with respect to risk of failure or resistance development

***Thank you for this comment. Because there is no previous experience with NVP- containing DT and in order to rapidly detect a possibly increased risk of viral failure, special safety precautions have been included in the protocol. First, a stopping rule was implemented (stop of the study in the event of viral failure in two patients) and second, the protocol required a stepwise inclusion (3 steps). This is additionally illustrated in the study flow (Figure 1). In order to further clarify this issue, we modified the sentence (Line 90-93), adding “in particular to prevent the development of resistance in the *event of virus failure”. All patients included signed informed consent containing information on the potential risk of failure and development of *resistance and this information was approved by the local ethical review board.***

4) The fact that NVP is not recommended in any high-income country guidelines should be acknowledged; similarly you need to provide some clinical context for recommending a regimen like this - why choose NVP/3TC over more established tripe/dual options?

***Thank you, that's correct. In order to make this clear, we have both changed the introduction (Line 61-64) and the conclusion (Line 205-206). Justification of this regimen is covered in the introduction as well as in the conclusion (more than 20 years of experience with documented long-term tolerance and efficacy over decades, compartment penetration, prize).***

5) Please include some discussion as to the value of pilots e.g. PADDLE is an example of promising pilot findings being replicated in a larger single-arm study and then large RCTs, PI/r + MVC is an example whereby promising single-arm results were not replicated in RCTs

***You're right, a successful pilot does not guarantee success in a large RCT. To disclose this concern, we have added an additional point to the limitations *(Line 191-193).***

---

## [Decision Letter · Decision Letter 1]

30 Jan 2020

PONE-D-19-28328R1

NOVEL DUAL HIV MAINTENANCE THERAPY WITH NEVIRAPINE PLUS LAMIVUDINE RETAIN VIRAL SUPPRESSION THROUGH 96 WEEKS - A PROOF-OF-CONCEPT STUDY

PLOS ONE

Dear Dr. Kahlert,

Thank you for submitting your manuscript to PLOS ONE. After careful consideration, we feel that it has merit but does not fully meet PLOS ONE’s publication criteria as it currently stands. Therefore, we invite you to submit a revised version of the manuscript that addresses the points raised during the review process.

Your above manuscript has been seen by two Referees. Only one referee made additional comments to the Authors. The report is included below.  Both referees believe that the work is of interest and can be accepted for publication if the criticisms they raised are properly addressed.

We would appreciate receiving your revised manuscript by the next 12 weeks. To enhance the reproducibility of your results, we recommend that if applicable you deposit your laboratory protocols in protocols.io, where a protocol can be assigned its own identifier (DOI) such that it can be cited independently in the future. For instructions see: http://journals.plos.org/plosone/s/submission-guidelines#loc-laboratory-protocols

We look forward to receiving your revised manuscript.

Kind regards,

Giuseppe Vittorio De Socio, MD, PhD

Academic Editor

PLOS ONE

Reviewers' comments:

Reviewer's Responses to Questions

**Comments to the Author**

1. If the authors have adequately addressed your comments raised in a previous round of review and you feel that this manuscript is now acceptable for publication, you may indicate that here to bypass the “Comments to the Author” section, enter your conflict of interest statement in the “Confidential to Editor” section, and submit your "Accept" recommendation.

Reviewer #2: All comments have been addressed

Reviewer #3: (No Response)

2. Is the manuscript technically sound, and do the data support the conclusions?

Reviewer #2: Yes

Reviewer #3: Partly

3. Has the statistical analysis been performed appropriately and rigorously? 

Reviewer #2: N/A

Reviewer #3: No

4. Have the authors made all data underlying the findings in their manuscript fully available?

Reviewer #2: Yes

Reviewer #3: No

5. Is the manuscript presented in an intelligible fashion and written in standard English?

Reviewer #2: Yes

Reviewer #3: Yes

6. Review Comments to the Author

Reviewer #2: Thank you for your amendments - you have addressed my comments thoroughly and I am happy to recommend the paper for publication.

Reviewer #3: This is a descriptive analysis of a small pilot study evaluating the potential use of the dual combination 3TC+NVP in people with current fully suppressed HIV-RNA who have shown previous long term virological success and tolerability on NVP-based triple therapy. The argument for testing this combination is the existence of over 20 years of use of these drugs in the clinics with documented long-term tolerance and efficacy, excellent penetration in sanctuary and reduced costs compared to other dual regimens (at least in Switzerland).

The main issue with these drugs in the context of dual therapy, rather than the risk of HSR, is the low barrier to drug resistance (not just for NVP but also for 3TC). The argument that a high penetration into sanctuaries may be more important than a high genetic barrier in the context of antiretroviral maintenance is not particularly convincing. This is because i) viral load response in this trial is shown as average proportions and ii) there is no measure of low-level HIV resistance in the study. More detailed results should be presented to deserve publication, I have some suggestions below:

Main points

1. What is the definition of 'fully suppressed' viral load (inclusion criteria of trial)? <1 copy/mL? <=50 copies/mL? The actual breakdown of the viral load distribution at week 0 could be shown in Table 1

2. Are there stored samples available in which to measure low-level drug resistance? It is currently possible to measure resistance even when viral load is very low. If authors could show that there was, say, no 103N or 184V in minor populations at week 96 this finding would decrease concerns around clinical use of this combination. HSR and cost are less important issues in my opinion. Most mono/dual combinations with low resistance barrier have shown inferiority when compared to standard triple regimens in head to head RCTs. The lack of resistance data should at least be mentioned in the Discussion.

3. With a sample size of 20 patients it is easy to show spaghetti plots with the full viral load trajectories for the all study population, including the 96 week before and after the inclusion into the pilot maintenance study. It is mentioned on line 190-192 that However, in this population, all patients kept full HIV pVL suppression and even the frequency of blips with DT was equal to the pre-study period where patients received cART with 3 compounds. This is not supported by data shown in Figure 2 because this includes average proportions and it is not possible to say whether blips where more/less frequent over the pilot study than over the previous period in individual patients. Indeed, there seems to be evidence in aggregate of people moving from the ≤1 copy/mL stratum (78% vs 86%) to the 2-20 copies/mL stratum (19% vs 10%) under dual therapy. Spaghetti plots will give more insights and possibility to develop specific statistical tests to compare the frequency of blips before and after the switch.

4. There is no control group in this trial so the only possible comparison is with historical values of viral load measured over the previous period under triple therapy. Intermittent time series approach for average viral load levels comparing the slope before and after the switch to dual can be used to confirm that there is no evidence for a difference in the slopes in the two periods. This auto-regressive method is very powerful and commonly used in mono-arm trials with no control group now also in HIV research.

5. Legend for Figure 2. I would modify the label for the stratum ‘>50’ into ‘50-100’ copies/mL. Or where there people who showed blip >100 copies/mL?

Other Points

1. Line 44-46. Afterwards, viral replication is completely suppressed and a simplified treatment with two antiretroviral compounds (dual therapy – DT) or even one substance (monotherapy) may control viral replication on the long run. This sentence needs to be qualified. None of the mono-therapy regimens have shown particularly encouraging results; same story applies to some of the dual combinations, shown to be inferior to triple therapy (e.g. those including maraviroc etc).

2. Line 171. Typo NVP and 3TC, not und

3. A previous reviewer pointed out that sentence in lines 233-234 was not particularly clear. This has now been revised in the current version in ‘In summary, given the sample size, our findings do not reject the null-hypothesis that dual therapy is equally effective as standard therapy’. I think that this is still inaccurate; the study does not support equivalence of the dual therapy with triple therapy because there is no control group. Lack of a plausible causal argument is more important than statistical power.

4. It is mentioned in the Conclusions (lines 237-238) and Abstract that ‘A properly sized multicentre non-inferiority trial is ongoing to further evaluate the value of this DT maintenance strategy’. Does this trial really exist? There seems to be no sign of such a trial in https://clinicaltrials.gov/. It would be useful to give more information on the status of such trial.

7. PLOS authors have the option to publish the peer review history of their article (what does this mean?). If published, this will include your full peer review and any attached files.

Reviewer #2: No

Reviewer #3: No

---

## [Editor Report · Decision Letter 2]

4 Aug 2020

NOVEL DUAL HIV MAINTENANCE THERAPY WITH NEVIRAPINE PLUS LAMIVUDINE RETAIN VIRAL SUPPRESSION THROUGH 144 WEEKS - A PROOF-OF-CONCEPT STUDY

PONE-D-19-28328R2

Dear Dr. Kahlert,

We’re pleased to inform you that your manuscript has been judged scientifically suitable for publication and will be formally accepted for publication once it meets all outstanding technical requirements.

Kind regards,

Giuseppe Vittorio De Socio, MD, PhD

Academic Editor

PLOS ONE
---

## [Editor Report · Acceptance letter]

9 Sep 2020

PONE-D-19-28328R2 

Novel Dual Hiv Maintenance Therapy With Nevirapine Plus Lamivudine Retain Viral Suppression Through 144 Weeks - A Proof-Of-Concept Study 

Dear Dr. Kahlert:

I'm pleased to inform you that your manuscript has been deemed suitable for publication in PLOS ONE. Congratulations! Your manuscript is now with our production department. 

Kind regards, 

on behalf of

Dr. Giuseppe Vittorio De Socio 

Academic Editor

PLOS ONE